# Application and Optimization of the Parameters of the High-Level Boreholes in Lateral High Drainage Roadway

Haiqing Shuang [1,2], Weitao Meng [1], Yulong Zhai [1,3,*], Peng Xiao [1,2], Yu Shi [1,2] and Yu Tian [1]

1 School of Safety Science and Engineering, Xi'an University of Science and Technology (XUST), No. 58, Yanta Mid. Rd., Xi'an 710054, China
2 Key Laboratory of Western Mine Exploitation and Hazard Prevention of the Ministry of Education, Xi'an University of Science and Technology (XUST), No. 58, Yanta Mid. Rd., Xi'an 710054, China
3 Department of Civil, Geological and Mining Engineering, École Polytechnique de Montréal, Montréal, QC H3C 3A7, Canada
* Correspondence: yulong.zhai@polymtl.com; Tel.: +86-139-9299-8589

**Abstract:** The key parameters of high-level boreholes in high drainage roadways affect the gas treatment effect of the working face directly. Therefore, the layout parameters of high-level boreholes in the lateral high drainage roadway (LHDR) are determined and optimized as necessary. Based on the LHDR layout on the 2-603 working face of the Liyazhuang coal mine, the key technological requirements on high-level borehole parameters were analyzed and the distribution characteristics of the gas volume fraction in the coal roof were studied. The gas migration law in the mined-out areas was obtained and the layout locations of high-level boreholes were determined finally. The research demonstrates that the high-level boreholes lag the 2-603 working face distance and the position of the final borehole (the position of the final borehole in this paper refers to the distance between the final borehole and the roof) influence the stability of boreholes and the gas extraction effect. The distribution of the gas volume fraction from the intake airway to the return airway can be divided into the stable stage, slow growth stage, and fast growth stage. Influenced by the flow field in the mined-out areas, the mean volume fraction of the borehole-extracted gas has no obvious relationship with the gas volume fraction at the upper corner. According to the final optimization, the high-level borehole is determined as having a 15 m lag behind the working face and the position of the final borehole is 44 m away from the coal seam roof. These have been applied successfully in engineering practice.

**Keywords:** lateral high drainage roadway; high-level boreholes; gas extraction; numerical simulation; flow field

## 1. Introduction

Coal bed methane (CBM) is a valuable resource and the disasters (e.g., gas explosions and coal-gas outbursts) affect the safety production in the process of coal mining [1–3]. The overlying rock of the working face will migrate and break after mining. The overlying strata will experience a significant number of fractures, providing pathways for gas storage and flow [4,5]. The gas reserve law in coal mines in China shows the feature of three high (high plasticity structure of coal seam, high gas adsorption ability of coal seam, and high gas content of coal seam) and three low (low gas pressure of coal seam, low gas saturation of coal seam, and low rate of gas permeability of coal seam), which causes the effect of pre-mining gas extraction to be ineffective. Many scholars combine the development characteristics of mining-induced fissures with the migration law of pressure relief gas [6–8] and use high drainage roadways and high-level boreholes to extract pressure relief gas [9–12].

Among them, Li et al. [13] adopted the technology of setting super large diameter (0.3 m) boreholes on the roof to ensure the safety production of high gas and outburst mines. Zhang et al. [14] selected the boreholes parameters reasonably through the observation



and trajectory measurement of gas drainage boreholes. Zhou et al. [15] conducted the theoretical analysis on the evolution law of mining-induced overlying strata fracture. The technique of gas extraction in different enrichment areas using different level boreholes was proposed. Shang et al. [16] proposed high directional long borehole drainage technology to control the gas concentration in the upper corner under the safety index of the mined-out areas. Li et al. [17] determined the key parameters of the (LHDR) by analyzing the mining stress. Yu et al. [18] studied the distribution and evolution characteristics of the stress and displacement field in the surrounding rock of the roadway and arranged the floor gas drainage roadway to control gas. Hu et al. [19] proposed a technique of a large-diameter blind shaft joined to the high-level gas drainage roadways to control gas. Many scholars have performed a detailed study on the gas extraction methods in the mined-out areas, among which the high-level boreholes and high drainage roadways are the most frequently employed. Fan [20] and Zhang [21] have also studied the key problems of arranging drilling holes for gas extraction.

Some scholars studied the gas extraction method (e.g., high-level boreholes extraction and high drainage roadway extraction) in the mine-out areas after mining [22,23]. Among them, high-level boreholes extraction is an efficient gas extraction method to control gas over the limit. However, the relationship between the lag working face distance and the boreholes stability is not clear. The influence law of the different layers of the final location of the high-level boreholes on the gas flow field migration is not clear. To study the law of gas flow field migration in the mined-out areas affected by the distance between high-level boreholes lagging working face and the position of the final borehole, UDEC$^{2D}$ was used to simulate the overburden fissures zone after mining and separation zone in the overlying strata of mined-out areas. The distance between high-level boreholes lagging working face and the final borehole position was determined. FLUENT was used to analyze the thermal characteristics of gas volume distribution in the coal seam roof and the distribution characteristics of the coal seam roof gas affected by mine-out areas flow field under the influence of drainage boreholes were obtained, to determine the drainage effect of different layers of high-level boreholes on the coal seam roof. The simulated high-level boreholes layout parameters were used to control gas over the limit in the upper corner of the Liyazhuang coal mine.

To prevent gas over-runs in the Liyazhuang coal mine, the pressure relief gas extraction technology (i.e., two adjacent working faces share the external staggered high drainage roadway) is proposed: vertical (vertical 2-603 working face) and lateral (vertical 2-605 working face) high drainage roadways are placed at the roof along the return airway on the 2-603 working face. In the early service stage of high extraction roadways, pressurized gas extraction technology was applied to the 2-603 working face through high-level boreholes. In the late service stage of high drainage roadways, pressurized gas extraction technology was used to the 2-605 working face through high drainage roadways directly. Based on the development characteristics of mining-induced overburden fissures and the reasonable position of high drainage roadways on the working face, the key position parameters of the final high-level borehole are analyzed to realize high-efficiency pressurized gas extraction in the overburden fissure zone of the 2-603 working face in the early service stage of LHDR.

## 2. Layout of LHDR

### 2.1. Determination of Key Parameters

The LHDR layout is proposed based on geological conditions and gas control measures in the Liyazhuang coal mine (Figure 1). Vertical and LHDR are placed at the roof along the return airway on the 2-603 working face. In the early service stage of high drainage roadways, pressurized gas extraction technology is applied to the 2-603 working face through high-level boreholes. In the late service stage of high drainage roadways, pressurized gas extraction technology is used to the 2-605 working face through high drainage roadways directly.

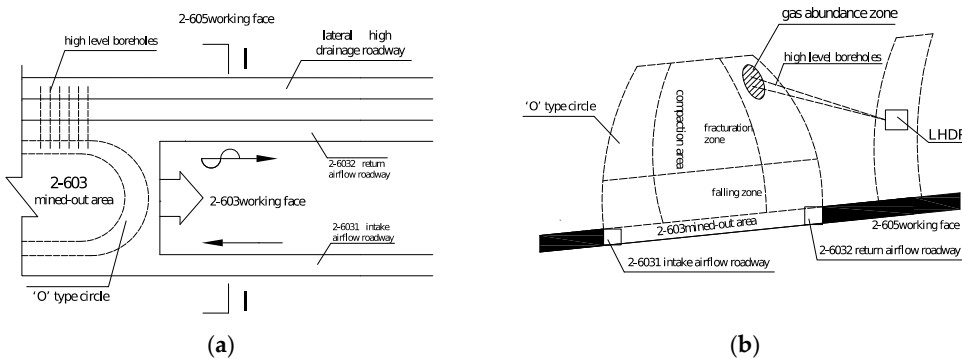

**Figure 1.** Surrounding rock structure mechanical model of high-level boreholes: (**a**) planar graph; (**b**) I-I section drawing.

The final borehole position directly affects the drainage effect of pressure relief gas in overlying rock mining fissures of the 2-603 working faces in the early stage. Because the high-level boreholes are arranged in the outer staggered high drainage roadway, which is affected by the mining of 2-603 working faces in the early stage, the layer position of the high drainage roadway directly affects the layout parameters of high-level boreholes. The position of the LHDR will influence the surrounding rock stress environment in the roadway directly. Therefore, the technological key of the LHDR layout is to determine parameters for the LHDR and the high-level borehole. Based on previous research results [17], the position parameters of LHDR are determined as follows: roof strike high drainage roadways are arranged along the 2-603 working face, which is 25 m away from the 2-603 working face horizontally and 25 m away from the 2# coal roof vertically. The position parameters of the high-level boreholes are analyzed in the following text.

*2.2. Requirements for High-Level Boreholes Parameters in LHDR*

From the layout of the LHDR, the high-level boreholes are mainly for pressurized gas extraction in mining-induced overburden fissures on the 2-603 working face. The position parameters of high-level boreholes are the key factors to early extraction pressure relief gas efficiency in the layout of the LHDR. Some requirements on the parameters of the high-level boreholes are provided below:

(1) Overburden mining-induced fissures determine the migration pathways and storage spaces for pressure relief gas directly. Therefore, it is necessary to master the distribution characteristics of overburden mining-induced fissures along the 2-603 working face. The position of final boreholes must be placed at the pressure relief gas enrichment region with consideration to pressure relief gas characteristics.

(2) Three horizontal areas (i.e., coal wall support area, separation area, and re-compaction area) will be formed along the advancing direction of the working face after mining. At the same time, three vertical zones (i.e., bending sagging zone, fracture zone, and caving zone) will be formed from the upper to the lower parts of the overburden areas [24,25], as shown in Figure 2.

From Figure 2, one sees that when the gas extraction boreholes are in the coal wall support area, the supporting stress for advanced mining of the working face will influence the borehole stability and cause the collapse or deformation of boreholes. When boreholes are in the separation area and re-compaction area, the mining-induced overburden fissures will become the gas migration pathways with pressure relief. The arrangement of boreholes in the re-compaction area slightly influences the pressure relief gas flow field on the working face and is easy to cause an exceeding gas concentration at the upper corner. The final high-level boreholes must be placed at the intersection of the fissure zone and separation area.

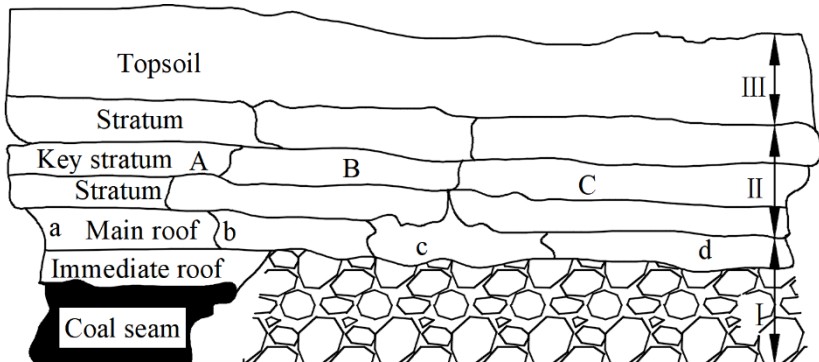

**Figure 2.** The horizontal zones and vertical zones after working face mining. (A—coal wall support area (a–b); B—separation area (b–c); C—re-compaction area (c–d); I—caved zone; II—fractured zone; III—bending zone).

(3) When high-level boreholes are behind the working face, the LHDR is disturbed by mining the 2-603 working face and the loosing zone of surrounding rocks close to the 2-603 working face increases accordingly. Therefore, the sealing length and strength of the final high-level boreholes must be increased to ensure negative pressure extraction and efficiency.

(4) The high-level boreholes are arranged in the common area of the fractured zone and the separation zone. It is necessary to ensure the stability of the high-level boreholes to achieve efficient gas extraction. The high-level boreholes should not be in the peak stress area [26]. It reduces the influence of mining stress on the stability of the boreholes and ensures the gas extraction effect of the boreholes.

To sum up, when high-level boreholes are behind the working face, the final boreholes must be in the separation area of the fracture zone, and the sealing strength of the high-level boreholes must be increased. At the same time, the mining stress will cause the boreholes to collapse when the working face continues to advance, so it is necessary to strengthen the sealing strength of high-level boreholes and the high-level boreholes should not be in the peak stress area. As a result, the high-level boreholes lag the working face distance as well as the position of the final borehole must be optimized to improve the boreholes stability and achieve a better extraction effect.

## 3. Distribution Characteristics of Mining-Induced Overburden Fracture Zone and Separation Area

### 3.1. Overburden Fissure Distribution Range

Figure 3 shows the distribution of the mining-induced fractures after mining. The mining-induced fissures distribution area of the overlying strata at the end of the working face is within 62° of the uphill mining angle. The fissures in the vertical direction are concentrated in two main areas: (1) the first area: 13–25 m away from the floor at the height, 12 m away from the stope boundary, and 65 m in width, and (2) the second area: 38.6–50 m away from the floor at the height, 28 m away from the stope boundary, and 50 m in width. The first place is closer to the working face, the air leakage is more serious. Since they are affected by the ascending characteristics of the gas, the extraction boreholes should be positioned above the second zone.

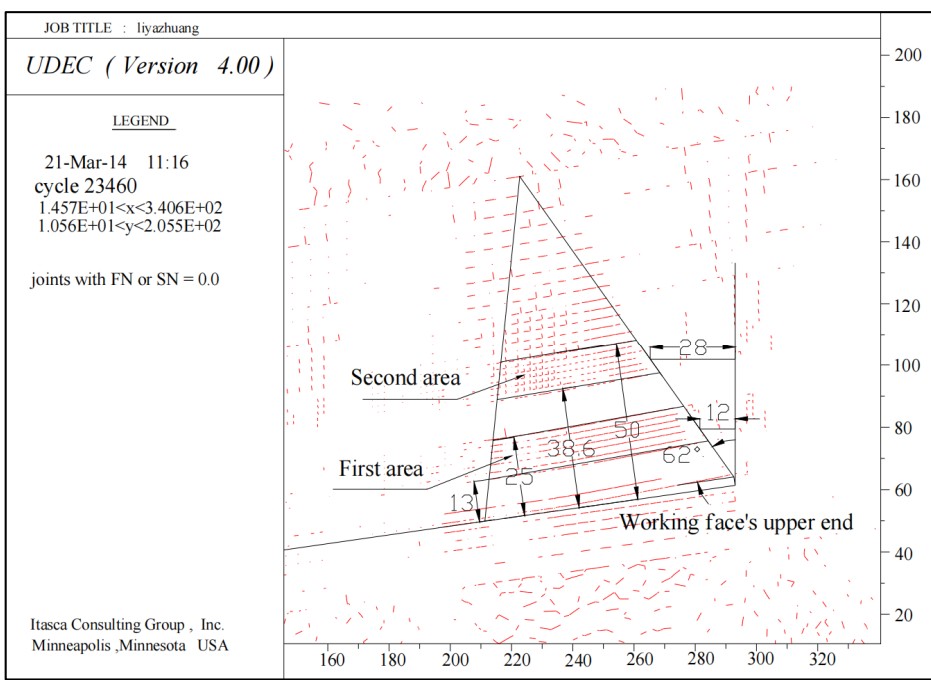

**Figure 3.** Distribution characteristics of overburden fissures after mining.

### 3.2. Distribution Characteristics of Separation Area

A numerical model was constructed according to the geological conditions of the 2-603 working face. The model parameters are shown in [17,27,28]. The distribution characteristics and extent of the mining overburden fissures after excavation of the 2-603 working face were simulated. The first pressurization and periodic pressurization occur with the advancing of the working face. The distribution characteristics of mining-induced overburden fissures when the working face advances by 95 m are shown in Figure 4.

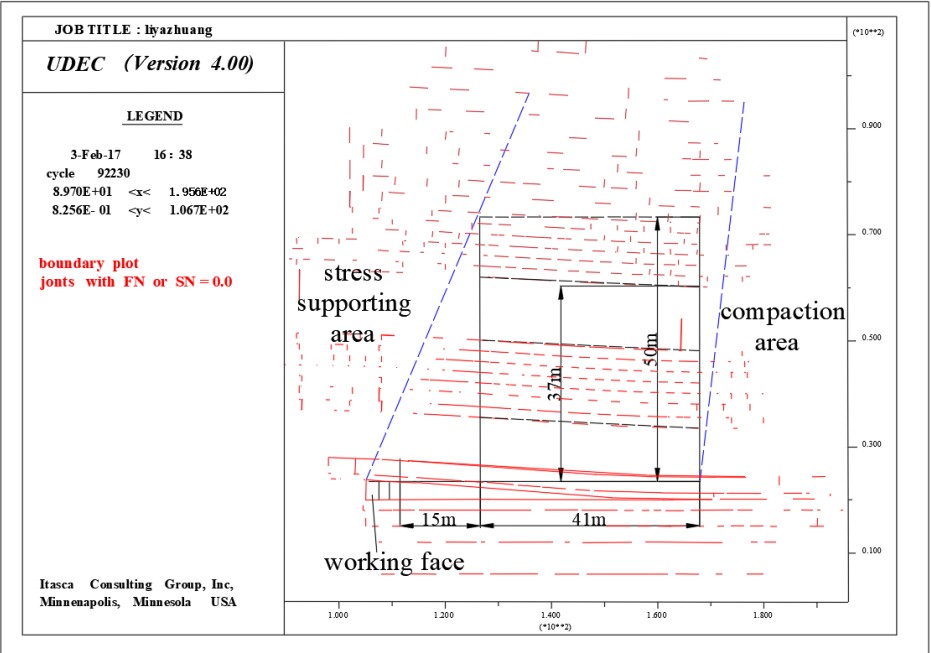

**Figure 4.** Distribution characteristics of mining-induced overburden fissures.

Figure 4 shows that the stress supporting area, separation area, and compaction area are developed on the working face after mining. There are few mining-induced overburden fissures in the stress supporting area and the compaction area, which limit pressure relief gas storage and migration. Since two layers of thick sandstones play a key role in controlling the overburden activities, the mining-induced fissures are mainly distributed in two areas, which are consistent with the distribution characteristics of mining-induced fissures on inclined overburden. The main storage pathways of pressure relief gas are mainly 15–41 m away from the working face horizontally and 37–50 m away from the roof vertically. Therefore, boreholes are set 15–41 m away from the working face and the final borehole is 37–50 m away from the roof.

## 4. Parameters of High-Level Boreholes Layout

### 4.1. Theoretical Basis of the Numerical Model

Gas flow in the mining-induced fracture zone is very complicated and can be simplified as a seepage in porous media, including the laminar region, transition region, and turbulence region. Generally, there is large air leakage in small mined-out areas range close to the working face, but air leakages in other regions are similar with seepage of a small Renault coefficient [29]. Hence, gas migration in the mining-induced fracture zone conforms to the continuity equation, momentum equation, and mass conservation equation of fluid flow in porous media [30,31].

(1) Equations of laminar flow and turbulence flow

Laminar flow and turbulence flow of pressure relief gas in the mining-induced overburden fissures on the working face using borehole extraction were designed. The $RNGk - \varepsilon$ model with high and low Reynolds numbers was applicable.

① Equation of turbulent kinetic energy

$$\frac{\partial}{\partial t}(\rho k) + \frac{\partial}{\partial x_i}(pku_i) = \frac{\partial}{\partial x}\left[a_k(u + u_l)\frac{\partial k}{\partial x_j}\right] + G_k + G_b - \rho\varepsilon - Y_M + S_K \qquad (1)$$

where $t$ is time (s); $\rho$ is air density (kg/m$^3$); $k$ is turbulent kinetic energy (m$^2$/s$^2$); $u_i$ is the temporal average velocity (m/s); $a_k$ is the number of turbulence flow; $u$ is the dynamic viscosity of the fluid (Pa $\cdot$ s); $u_l$ is turbulence viscosity (Pa $\cdot$ s); $G_b = -g_i\frac{u_t}{\rho pr_t}\frac{\partial \rho}{\partial x_i}$ ($g_i$ is the acceleration of gravity (m/s$^2$) and $pr_t$ is *Prandtl* number of turbulence); $Y_M = 2\rho\varepsilon M_t$ ($M_t$ is Mach number of turbulence, $M_t = \sqrt{k/a^2}$ and $\alpha$ is the sound velocity (m/s )); $\varepsilon$ is dissipation rating (m$^2$/s$^3$); $G_K = u_l\left(\frac{\partial u_i}{\partial x_i} + \frac{\partial u_j}{\partial x_i}\right)\frac{\partial u_i}{\partial x_j}$; and $S_K$ is the source item (kg/ (m $\cdot$ s$^3$)).

② Diffusion equation of the dissipation rating of turbulent kinetic energy ($\varepsilon$)

$$\frac{\partial}{\partial t}(\rho\varepsilon) + \frac{\partial}{\partial x_i}(\rho\varepsilon u_i) = \frac{\partial}{\partial x_i}\left[a_\varepsilon(u + u_t)\frac{\partial\varepsilon}{\partial x_i}\right] + C_{1\varepsilon}\frac{\varepsilon}{k}(G_k + C_v G_b) - C_{2\varepsilon}\rho\frac{\varepsilon^2}{k} - R_\varepsilon + S_\varepsilon \qquad (2)$$

where $a_\varepsilon$ is the *Prandtl* number of turbulence; $C_{1\varepsilon}$, $C_{2\varepsilon}$ and $C_v$ are dimensionless constants of the model; $R_\varepsilon = \frac{0.0845\rho\eta^3\left(1 - \frac{\eta}{4.377}\right)}{1 + 0.012\eta^3} \cdot \frac{\varepsilon^3}{k}$; kg/ (m $\cdot$ s$^4$); $\eta = (2E_{ij} \cdot E_{ij})^{1/2}\frac{k}{\varepsilon}$; $E_{ij} = \frac{1}{2}\left(\frac{\partial u_i}{\partial x_j} + \frac{\partial u_j}{\partial x_i}\right)$, s$^{-1}$; and $S_\varepsilon$ is the source item (kg/ (m $\cdot$ s$^4$)).

Moreover, the model equation of effective velocity and eddy modification gram mark was involved to adapt to the influences of the eddy on low Reynolds number, near-wall flow, and modified turbulence flow in the laminar flow:

$$d\left(\frac{\rho^2 k}{\sqrt{u_t\varepsilon}}\right) = 1.72\frac{\hat{v}}{\sqrt{\hat{v}^3 - 1 + C_v}}d\hat{v} \qquad (3)$$

where $\hat{v} = (u + u_t)/u$; $u_t$ differs for different liquid states. Turbulence flow and eddy are determined according to Equation (4).

$$u_l = 0.0845\rho k^2/\varepsilon u_t = u_{t_0} f(a_s, \Omega, k/\varepsilon) \tag{4}$$

where $u_{t_0}$ is a non-revised value of viscosity (Pa · s); $a_s$ is constant; $\Omega$ is the estimated value.

(2) Mass conservation equation of gas

In the mining-induced fracture zone, gas transmission in the air must meet obey the mass conservation law.

$$\frac{\partial}{\partial t}(\rho c_g) + \frac{\partial}{\partial x_i}(\rho c_g u_i) = -\frac{\partial}{\partial x_i}(J_g) + S_g \tag{5}$$

where $c_g$ is the volume fraction of gas, $m^3/m^3$; $J_g$ is gas diffusion flux (kg/(m$^2 \cdot$ s)); laminar flow and turbulence flow are determined by Equation (6). $S_g$ is the additional productivity of the gas source item ($\frac{\text{kg}}{(\text{m}^3 \cdot \text{s})}$).

$$J_g = -D\rho \frac{\partial}{\partial x_i}(c_g), J_g = -\left(D\rho + \frac{u_t}{Sc_t}\right)\frac{\partial}{\partial x_i}(c_g) \tag{6}$$

where $D$ is the diffusion coefficient of gas (m$^2$/s) and $Sc_t$ is Schmidt number of turbulence flow.

(3) Equation of continuity

According to the mass conservation law of gas in mining-induced fissures, the equation of continuity of gas migration is gained:

$$\frac{\partial \rho}{\partial t} + \frac{\partial}{\partial x_i}(\rho u_i) = S_g \tag{7}$$

where $S_g$ is the gas source item (kg/(m$^3 \cdot$ s)).

(4) Momentum conservation equation

The momentum conservation equation of porous medium on the direction $i$ in the inertial coordinate system is:

$$\frac{\partial}{\partial t}(\rho u_i) + \frac{\partial}{\partial x_j}(\rho u_i u_j) = \frac{\partial \tau_{ij}}{\partial x_j} - \frac{\partial p}{\partial x_i} + \rho g_i + \sum_{j=1}^{3} D_{ij}(\mu + \mu_i)q_j + \sum_{j=1}^{3} C_{ij}\frac{1}{2}\rho|q_j|q_j \tag{8}$$

where P is the pore pressure (Pa); $q_j$ is the seepage velocity (m/s); $D_{ij}$ and $C_{ij}$ are specified matrixes; $\tau_{ij}$ is the stress tensor (Pa); $\tau_{ij} = (\mu + \mu_i)\left[\left(\frac{\partial u_i}{\partial x_j} + \frac{\partial u_j}{\partial x_i}\right) - \frac{2}{3}\frac{\partial u_i}{\partial x_i}\delta_{ij}\right]$ and $\delta_{ij}$ is the Kronecker mark.

Equations (1), (2), (5), (7) and (8) form the mathematical model of gas migration in the mining-induced fracture zone. If variables, time-varying terms, convective terms, and diffusion terms in all equations are expressed in standard forms, the general forms could be gained (Equation (9)). In the application, only the solver of Equation (9) is programmed, thus enabling to solve gas migration in the mined-out areas under different boundary conditions.

$$\frac{\partial}{\partial t}(\rho\phi) + div(\rho\phi u) = div(\Gamma grad\phi) + S \tag{9}$$

where $\phi$ is the universal variable; $\Gamma$ is the generalized diffusion coefficient and $S$ is the generalized source item.

*4.2. Parameter Setting*

The permeability of the coal seam and overlying rock is the main factor that controls gas outburst on the working face. The mining-induced stress distribution influences the permeability of both the exploitation bed and the adjacent bed. The permeability is

determined by the fissure development in the strata in front of the working face as well as the stress relief behind the working face. The stress distribution caused by mining influenced the permeability of both mining and adjacent layers. Based on analysis of stress distribution law in the goaf and previous experiences on FLUENT simulation study, permeability distribution in the goaf was determined. Permeability changes in different regions were $10^{-4}$–$10^{-6}$ m$^2$ and the maximum viscosity permeability in the goaf was about $10^{-10}$ m$^2$ [17,32].

Other parameters of the model are: intake airway is $5 \times 3$ m; the net fracture surface is 15 m$^2$; return airway is $4.8 \times 3.8$ m; the net fracture surface is 18.24 m$^2$. The inclined length, width, and height of the mining working face are 200 m, 5 m, and 3.5 m, respectively. The dip angle is 8°. The strike length and height of goaf are 250 m and 100 m, respectively. The model uses a local grid subdivision and the grid quantity of the whole model is about 1.12 million. The reference pressure of the model is set at 101.325 KPa. The simulated calculating pressure is the relative pressure value. The inlet of the intake airway is set free and the outlet of the return airway is set as a pressure outlet. The negative pressure ventilation is employed. The extraction borehole is pressure out and the negative pressure for extraction is 30 KPa.

*4.3. Scheme Settings*

To verify the reasonability of the established model, the working face open-off cut when no gas extraction in the mined-out areas and the distribution characteristics of gas volume fraction in the return airway were analyzed (Figure 5).

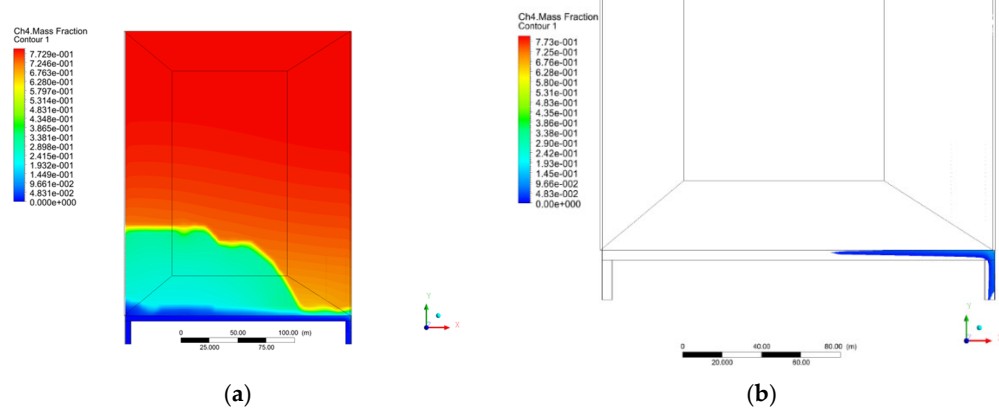

(**a**)          (**b**)

**Figure 5.** Gas distribution characteristics on the working face without extraction: (**a**) Contour of gas volume fraction at the roof; (**b**) Regions with gas volume fraction>1% on the working face.

Distribution characteristics of gas volume fraction (>1%) at the upper corner of the working face under the ventilation conditions are shown in Figure 5b. This region is mainly in the position that is 180–200 m away from the intake airway. The gas volume fraction at the joint between the working face and the return airway reaches 1.45–2.10% and the gas volume fraction on the working face exceeds the limit. Furthermore, practices have proved that increasing the ventilation quantity alone is difficult to solve excessive gas volume fraction at the upper corner of the working face. On one hand, increasing the ventilation quantity will discharge the gas into the working face and roadways. On the other hand, it will intensify air leakage in the goaf and more gases in the goaf will rush into the upper corner of the working face. Therefore, borehole gas extraction in the goaf must be considered when increasing the ventilation quantity fails to improve gas extraction. Combined with practical distribution characteristics of gas volume fraction on the working face, the established model is proved reasonable.

Combined with the distribution characteristics of mining-induced overburden fissures on the working face in Section 2.1, the periodic weighting step on the working face is about 15 m and the angle of mining-induced fissures at end of the 2-603 working face is 62°.

High-level boreholes are placed behind the working face and the experimental scheme is presented in Table 1. According to the line pressure for extraction in LHDR, 20 high-level boreholes were set at an interval of 2 m.

**Table 1.** Experimental Scheme of the different positions of the final boreholes.

| Distance from Working Face/m | 35 m away from Coal Seam Roof | 45 m away from Coal Seam Roof | 55 m away from Coal Seam Roof |
|---|---|---|---|
| 10 | Scheme1 | Scheme2 | Scheme3 |
| 15 | Scheme4 | Scheme5 | Scheme6 |
| 20 | Scheme7 | Scheme8 | Scheme9 |

*4.4. Analysis of Simulation Results*

Influenced by floating diffusion, pressure relief gas mostly concentrates in the top of the fissure region. Distribution characteristics of gas volume fraction close to the roof can represent the pressurized gas extraction effect of high-level boreholes to some extent. In the following text, reasonable parameters of high-level boreholes are analyzed from the distribution characteristics of gas volume fraction on the roof, the distribution law of gas volume fraction on the working face, and the average extracted gas volume fraction of high-level boreholes.

1.  Distribution characteristics of gas volume fraction on the roof;

Figure 6 shows that the distribution characteristics of gas volume fraction on the roof vary for high-level boreholes located at different positions. The region with low gas volume fraction close to the working face is A area (1.30% > gas volume fraction > 0) and the region with high gas volume fraction close to the working face is B area (gas volume fraction > 1.30%) (Figure 6a).

When boreholes are 10 m away from the 2-603 working face, the A area increases significantly with an increase in the height of the final borehole. B area expands to the return airway gradually, indicating the flow field close to the roof is beneficial for reducing gas discharge on the working face when the final borehole is 55 m away from the working face. When boreholes are 15 m away from the working face, the area of A decreases gradually with the increase in height of the final borehole, while the area of B remains the same. In addition, the B area gradually moves away from the return airway and develops to the deep part of the mining area on the side of the intake airway; when the boreholes lag behind the working face by 20 m, the area of the A area gradually decreases with the rise of the final hole position of the high drilling hole, and the B area gradually moves away from the return airway and moves to the deep part of the mining area on the side of the itay.

2.  Distribution characteristics of gas volume fraction on the roof;

The distribution law of gas volume fraction on the side of the coal seam roof will change under the influence of the mined-out areas flow field after extraction, which is reflected by the distribution characteristics of gas volume fraction on the coal seam roof. The distribution law is the macroscopic display of the flow field in the goaf. The distribution characteristics of gas volume fraction on the working face are closely related to this distribution law. To reflect the distribution law of gas volume fraction on the working face intuitively, the monitoring line of gas volume fraction is set between the intake airway and return airway, which is 3.0 m away from the bottom board of the coal bed. Distribution laws of gas volume fraction on the working face under different schemes are presented in Figure 7.

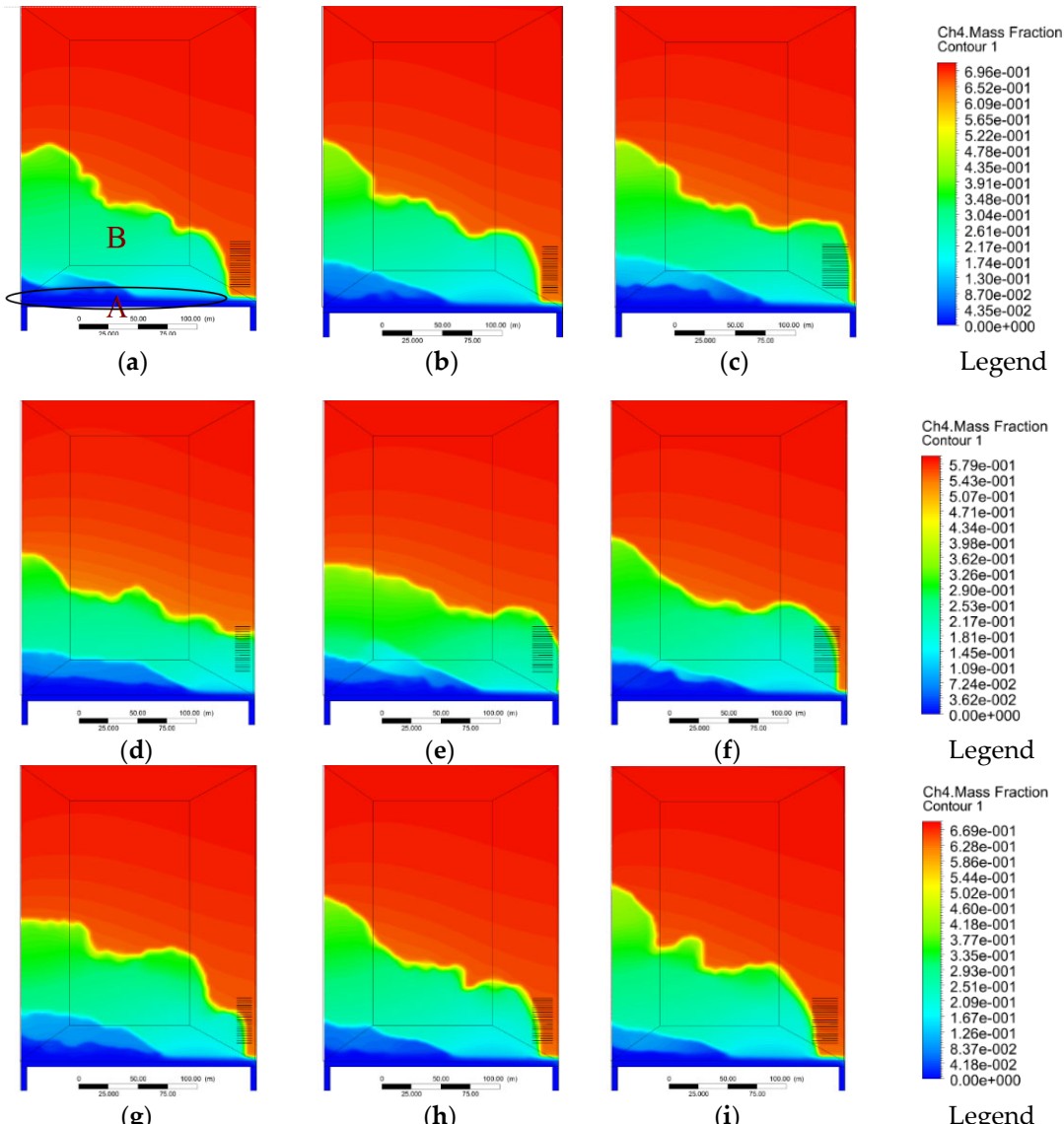

**Figure 6.** Distribution characteristics of gas volume fraction in different schemes: (**a**) scheme1; (**b**) scheme2; (**c**) scheme3; (**d**) scheme4; (**e**) scheme5; (**f**) scheme6; (**g**) scheme7; (**h**) scheme8; (**i**) scheme9.

In Figure 7, the gas volume fraction at the upper corner under extraction schemes decreases significantly compare with non-extraction schemes. The maximum gas volume fraction at the upper corner is 2.2% for no extraction scheme. For the borehole extraction, Schemes 3, 6, and 9 are over 1% gas volume fraction at the upper corner and the maximum gas volume fraction is 1.5%. This reflects that the borehole extraction can change the flow field for gas migration in the goaf and change distribution characteristics of gas volume fraction on the working face effectively.

The gas volume fraction on the working face is positively correlated with the distance increase from the position of the final borehole to the inlet airway. The growth of gas volume fraction is divided into three stages: the stable stage, slow growth stage, and fast growth stage. The variation laws of gas volume fraction on the working face under non-extraction and borehole extraction are different. Under the non-extraction technique, three stages are 0–26 m, 26–180 m, and 180–200 m in the intake airway, and the corresponding variation ranges are 0.0–0.012%, 0.012–1.00%, and 1.00–2.10%, respectively. Under borehole extraction technique, three stages are 0–110 m, 110–180 m, and 180–200 m, and the corresponding variation ranges are 0–0.10%, 0.10–0.41%, and 0.41–1.49%, respectively.

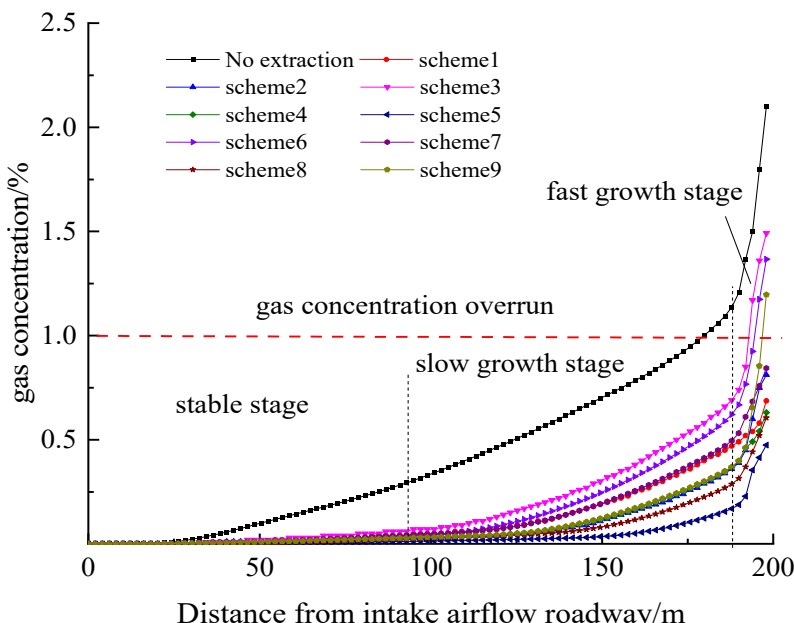

**Figure 7.** Distribution of gas volume fraction in mining working face.

Figure 7 shows that position parameters of boreholes influence the distribution law of gas volume fraction on the working face directly. According to schemes 1, 4, and 7, boreholes at the lower part of the fracture zone are conducive to gas extraction when the final borehole is 35 m away from the roof. The mining-induced overburden fissures are the main pathways for migration of pressure relief gas rather than the concentration area of pressure relief gas. The maximum gas volume fraction at the upper corner is 0.68%, 0.63%, and 0.84%, respectively. For schemes 2, 5, and 8, boreholes at top of the fracture zone can effectively extract pressure relief gas accumulated in this area when the final borehole is 45 m away from the roof. The maximum gas volume fraction at the upper corner is 0.81%, 0.47%, and 0.61%, respectively. It can be known from schemes 3, 6, and 9 that the borehole negative pressure cannot guide the gas flow field effectively when boreholes are 55 m away from the roof and the pressure relief gas volume fraction at the lower part rather than migrating to the 55 m, thus resulting in excessive gas volume fraction on the working face. The maximum gas volume fraction at the upper corner is 1.49%, 1.37%, and 1.20%, respectively.

3. Average drainage gas volume fraction of high-level boreholes and gas volume fraction at the upper corner.

The gas volume fraction of high-level boreholes influences the late use and extraction effect of gas directly. The gas volume fraction at the upper corner of the working face can reflect the gas control efficiency on the working face directly. Hence, position parameters of high-level boreholes for gas extraction must be determined by combining distribution characteristics of gas volume fraction on the roof and the distribution law of gas volume fraction on the working face. The gas volume fraction of high-level boreholes and gas volume fraction at the upper corner of the working face is shown in Figure 8.

Figure 8 shows that when the position of the final borehole is at different positions, the average drainage gas volume fraction of the high-level boreholes is different from that at the upper corner of the working face. The gas volume fractions at the upper corner in schemes 3, 6, and 9 exceed the limit (the safety concentration is below 1% at the upper corner), while those in schemes 2 and 7 are close to the safety warning line (0.81% and 0.84%, respectively), which are neglected in this paper. Only schemes 1, 4, 5, and 8 are feasible. The corresponding gas volume fraction at the upper corner is 0.69%, 0.63%, 0.47% and 0.61%, and the average drainage gas volume fraction of boreholes are 66%, 60%, 64%, and 55%, respectively. Therefore, the maximum variations of gas volume fraction at the upper corner and the average drainage gas volume fraction of boreholes in these 4 schemes

are 0.22% and 11.00%, respectively. The gas volume fraction of borehole extraction changes slightly. Combined with Figure 6, scheme 1 achieves the highest gas volume fraction at the same position 120 m behind the intake airway, followed by scheme 4, scheme 8, and scheme5 successively. This reveals that scheme 5 is more reasonable compared to the rest schemes.

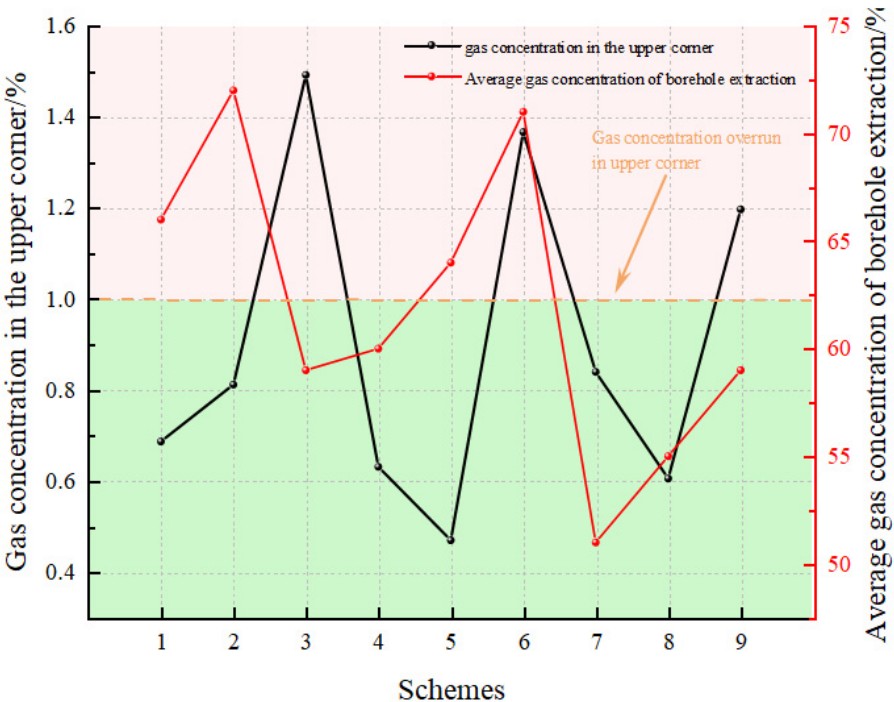

**Figure 8.** Average gas concentration of drainage borehole and gas concentration in the upper corner.

To sum up, gas extraction effect of high high-level boreholes represented by scheme 5 is best. In other words, when boreholes are 15 m away from the working face and the final boreholes are 45 m away from the coal roof, the average drainage gas concentration of boreholes is 64.00% and the gas volume fraction at the upper corner of the working face is within the limit, meeting the gas control requirements of the working face.

## 5. Engineering Application

### 5.1. Layout Parameters of Test Boreholes and Effect Analysis

1. Layout parameters of test boreholes:

To determine the reasonable position of the final borehole, the final borehole will be 15.0 m away from the working face and six boreholes are placed in the sector pattern close to the 115# borehole. The borehole diameter is 0.113 m and the interval of boreholes is set to 5.0 m. The construction position of boreholes are 1.0 m from roadway floor. The detailed borehole parameters and fissures distribution are shown in Figure 9.

The high-level boreholes are placed behind the working face. Based on the distribution characteristics of the mining-induced overburden fissures, the boreholes are connected with the mining-induced overburden fissures. Each borehole is sealed immediately after finishing the construction. Self-plugging pocket sealing technology (it uses polyurethane/polyurea composite material) is applied and the sealing length is determined as 10 m [26,33]. At this moment, high-level boreholes are mainly distributed in the separation area. The LHDR and borehole walls are affected by mining-induced stress on the working face slightly. The polyurethane/polyurea composites material is adequate to meet the sealing requirements. However, the distance from the external obstruction of the metal casing pipe to the internal mouth must be no less than 0.5 m to avoid air leakage caused by the connection between boreholes and the surrounding rock loose zone in the LHDR.

The orifice plate flowmeter and tailrace were connected outside the metal casing pipe. The buried pipelines were connected with the branching units on the gas drainage pipe through the air outlet of the tailrace.

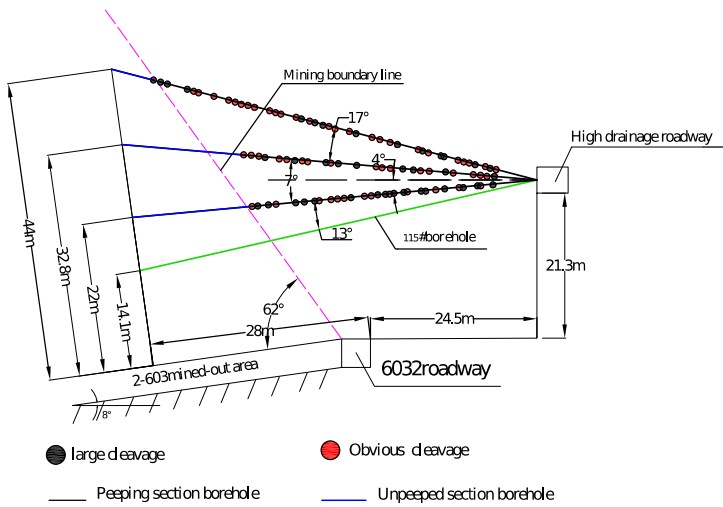

**Figure 9.** Specific parameters of test boreholes.

2. Sealing technology and key parameters of test boreholes:

To improve the connectivity and sealing quality of the high-level boreholes, the ponding and rock debris in the boreholes must be cleaned before the sealing. Due to timely sealing, the water on the borehole wall or mining-induced overburden fissures have not been eliminated in time, thus requiring irregular water drainage during the borehole extraction and protecting the high-level borehole efficient gas extraction of the pressurized gas.

3. Analysis of test borehole extraction effect:

The monitoring results of the gas volume fraction of six testing boreholes are shown in Figure 10. The effective gas extraction days of the boreholes 1-1 and 1-2 are 42 days and 39 days, respectively, with the corresponding average gas volume fractions of 52.4% and 43.0%, respectively. There are 12 and 13 days with gas drainage concentrations higher than 80%. The effective gas extraction days of the boreholes 2-1 and 2-2 are 23 days and 36 days and the corresponding average gas volume fraction is 12.6% and 17.8%, respectively. The gas drainage concentration mainly ranges between 10% and 30%, lasting for 17 days and 25 days. The effective gas extraction days of the boreholes 3-1 and 3-2 are 26 days and 32 days and the corresponding average gas volume fraction is 11.0% and 8.0%, respectively. The gas drainage concentration mainly ranges between 10% and 30%, lasting for 17 days and 21 days. To sum up, the position parameters of boreholes 1-1 and 1-2 are reasonable, which contribute to the large gas drainage concentration of boreholes and the long duration. They are capable of meeting the requirements for gas extraction in high-level boreholes.

4. Stability analysis of test borehole:

To analyze the borehole fissure morphology, the theoretical design borehole trajectory is expressed in two dimensions, the distance from the working face to the drilling field is represented by the horizontal axis, and the distance from the boreholes to the coal roof is represented by the vertical axis; the graphics represent the development of borehole fracture. The original fracture is represented by $\otimes$ and the new fracture is represented by $\odot$, where the bubble size is the crack size. Among them, the bubble size of 0.20 represents small fracture, 0.35 represents soft rock, 0.45 represents obvious fracture, 0.60 represents small-scale hole cutting, 0.65 represents small-scale hole collapse, 0.75 represents large-scale hole collapse, 0.80 represents complete hole cutting, and 1.00 represents borehole bottom. The fracture morphology of each borehole is shown in Figure 11.

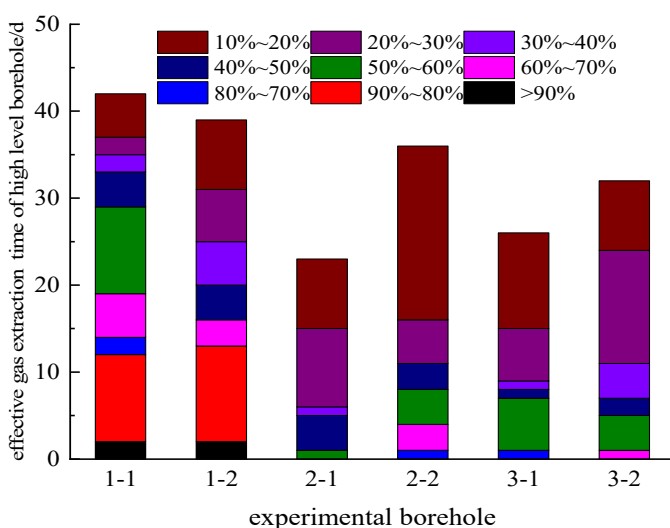

**Figure 10.** Test boreholes extraction gas volume fraction characteristics.

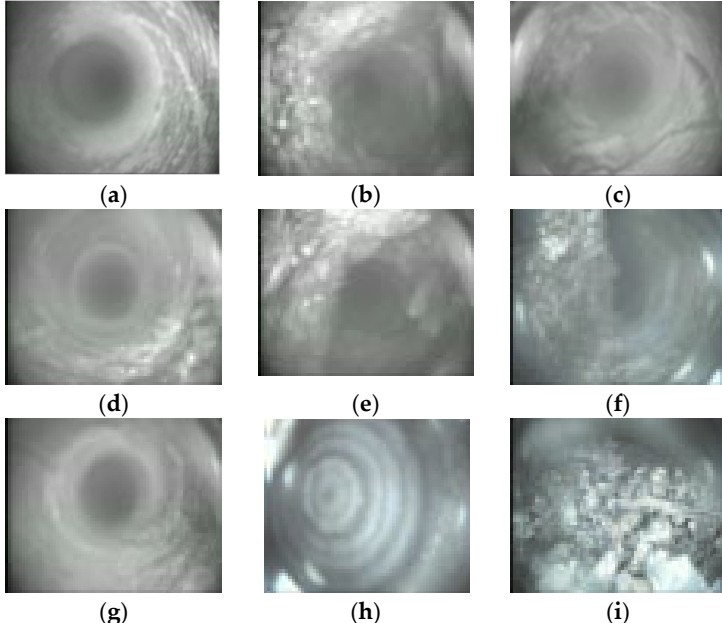

**Figure 11.** Classification of borehole observation fissures: (**a**) Small fissure; (**b**) Soft rock; (**c**) Obvious fissure; (**d**) Small-scale hole cutting; (**e**) Small-scale hole collapse; (**f**) Large-scale hole collapse; (**g**) Complete hole cutting; (**h**) Borehole bottom; (**i**) Complete hole collapse.

According to the analysis of the gas volume fraction characteristics of the test boreholes, it is known that the drainage effect of boreholes 1-1 and 1-2 are the best. According to the fracture evolution characteristics of the above two boreholes, the above two boreholes are analyzed in stages when the working face is advanced by 15 m and 30 m.

(1)    Working face advancing 15 m:

Figure 12 shows that there is a slight hole collapse phenomenon in the borehole between 12 m and 18 m. The fracture in the other borehole section is unevenly released and the borehole forming quality is poor. There is no fracture development and obvious deformation and the bottom of the hole is complete because the borehole has not received the influence of mining pressure at this time. Because the 2-603 working face is far away from the drilling sites, the end of the borehole has not yet fallen the square of the mined-out area. Combined with the borehole observation, it can be seen that the microfracture in the

hole section is mainly caused by the damage in the borehole forming process, borehole disturbance, water immersion, and the influence of pre-mining abutment pressure.

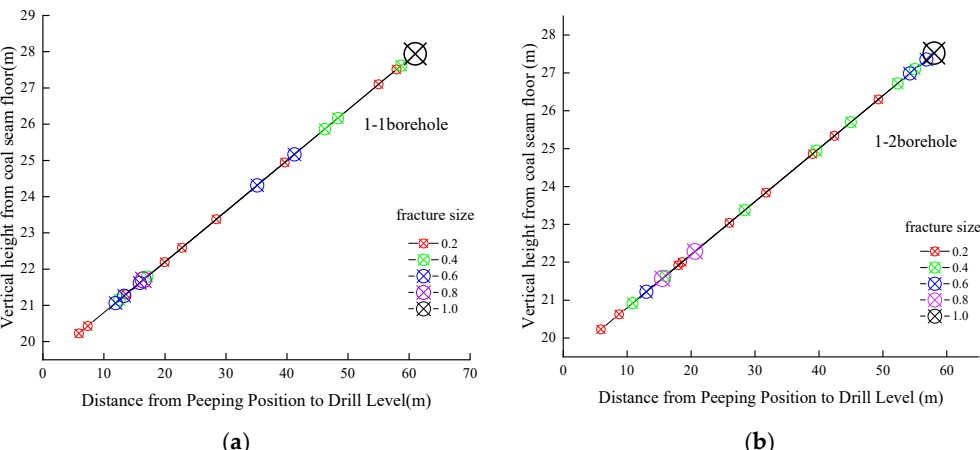

**Figure 12.** Distribution characteristics of borehole fissures when the working face is advanced 15 m: (**a**) Distribution characteristics of fissures in No.1-1 borehole; (**b**) Distribution characteristics of fissures in No.1-2 borehole.

(2)　Working face advancing 30 m:

Figure 13 shows that the final high-level borehole position is directly above the mined-out area at this time and the fracture development of each borehole has changed significantly. The fractures of the No. 1-1 borehole are dense at 24–36 m, they are obvious fractures and hole collapse, and there are cuts at the end of the borehole, which indicates that the end of the borehole is in the fracture zone; the situation of the No.1-2 borehole is similar to that of borehole 1-1. The results show that the high-level borehole stability is enough to ensure the extraction effect.

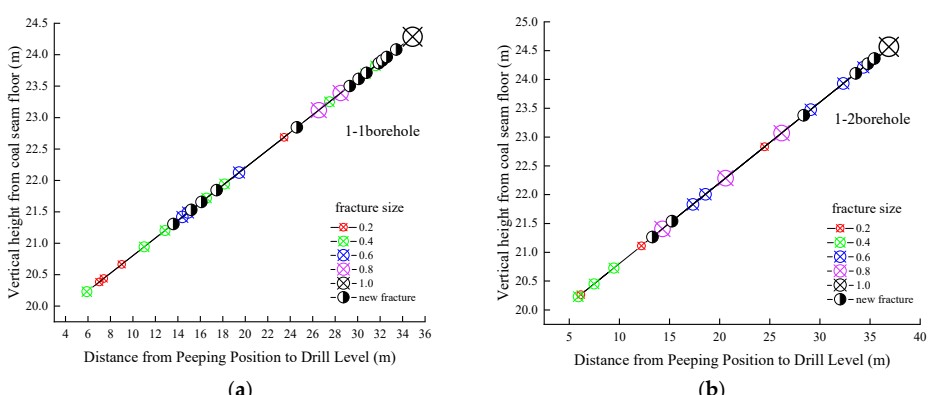

**Figure 13.** Distribution characteristics of borehole fractures when the working face is advanced 30 m: (**a**) Distribution characteristics of fissures in the No.1-1 borehole; (**b**) Distribution characteristics of fissures in No.1-2 borehole.

### 5.2. Late Engineering Applications

1.　Borehole layout parameters:

The extraction boreholes are arranged 15 m behind the 2-603 working face with a spacing of 1.8 m and an orifice of 1.0 m from the LHDR bottom board. The distance is about 17–29 m from the bottom of the high drainage roadway to the working face. The extraction drilling angle should be adjusted in real time to ensure the final borehole is 44 m above the roof of the 2-603 working face and the projection length of the borehole in the goaf is at least 28 m.

2. Layout parameters of test boreholes:

During the mining at the 2-603 working face, the gas volume fractions of the branch pipelines, LHDR boreholes, and the upper corner of the working face are monitored daily. The results show that the extraction branch is connected with from 15 to 20 boreholes, the extraction time can reach 20–40 days, and the gas volume fraction varies between 10% and 65%. The average flow rate of the pure gas is 22.3 $m^3$/min. The gas concentrations varied from 0.50–0.95% for the production shift and 0.47–0.89% for the maintenance shifts, respectively.

## 6. Conclusions

1. Key technological requirements for the high-level borehole layout are proposed based on the early service of the high-level boreholes in the layout of the two adjacent working faces sharing the LHDR. Specifically, the high-level boreholes are placed behind the working face and the position of the final borehole is in the overburden fracture zone and the separation area. The sealing strength of the gas extraction boreholes must be enhanced.
2. Gas migration in the mining-induced fracture zone is composed of equations of laminar flow and turbulence flow, continuity equation, momentum equation, and mass conservation equation. The FLUENT numerical simulation model is established according to the distribution law of the mining-induced overburden fissures on the 2-603 working face.
3. The layout parameters of high-level boreholes influence the gas distribution characteristics in the mined-out area directly, thus resulting in different gas migration characteristics on the roof, different gas volume fraction distribution patterns on the working face, and different average drainage gas volume fractions of high-level boreholes in nine schemes. According to the test results on the gas drainage concentration of boreholes, it concludes that the high-level boreholes must be 15.0 m away from the 2-603 working face and the final borehole must be 45 m away from the roof.
4. In engineering practice, the high-level boreholes are behind 15 m away from the working face and 15–20 boreholes are placed (enhancing sealing strength) at an interval of 1.8 m. The final borehole is located at 44.0 m away from the coal roof. According to the application effects, the average flow rate of pure gas is 22.3 $m^3$/min, which controls the gas volume fraction at the upper corner within the limit and thereby ensures safety and high-efficiency mine of the working face.

**Author Contributions:** Conceptualization, H.S.; software, H.S. and W.M.; data curation, W.M.; formal analysis, P.X., H.S. and Y.Z.; funding acquisition, Y.S.; investigation Y.T. and W.M.; writing—original draft, W.M.; writing—review and editing, Y.T., H.S. and Y.Z. All authors have read and agreed to the published version of the manuscript.

**Funding:** This research was funded by the National Natural Science Foundation of China (no. 51904238, no. 11802231, no. 51774235), China Postdoctoral Science Foundation (2019M663937XB), Shaanxi Natural Science Youth Foundation (no. 2019JQ-337), and Special Scientific Research Plan of Shaanxi Provincial Department of Education (no. 19JK0534).

**Institutional Review Board Statement:** Not applicable.

**Informed Consent Statement:** Not applicable.

**Data Availability Statement:** Not applicable.

**Acknowledgments:** This work is supported by the National Natural Science Fund Project (no. 51904238, no. 11802231, no. 51774235), China Postdoctoral Science Fund (2019M663937XB), Shaanxi Natural Science Youth Fund (no. 2019JQ-337), Special Scientific Research Plan of Shaanxi Provincial Department of Education (no. 19JK0534).

**Conflicts of Interest:** The authors declare no conflict of interest.

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
