# Peer review of "Application and Optimization of the Parameters of the High-Level Boreholes in Lateral High Drainage Roadway"

_sustainability, doi:10.3390/su142416908_

Round 1
Reviewer 1 Report
The article is interesting and topical especially now that we are facing a gas crisis. The presentation and treatment of the problem is clear and concise. The extraction of methane gas from the mined space is a known and currently applied method in mining. The way of treatment in the paper regarding the establishment on scientific bases of the geometry of the layout of the boreholes is interesting.
There are two issues to be aware of:
1- The depression applied to the boreholes must be correlated with the accumulated gas flow because there is a risk of self-ignition of the coal lost in the exploited area;
2- The boreholes must be secured with metal pipes and sealed both to increase the gas extraction efficiency and to avoid spontaneous combustion phenomena in the area of the boreholes.
Author Response
1- The depression applied to the boreholes must be correlated with the accumulated gas flow because there is a risk of self-ignition of the coal lost in the exploited area;
Response: Thank you for the reviewer’s very professional and leading-edge comment, according to the air distribution volume of the working face and engineering practice in the Liyazhuang coal mine. The depression applied to the boreholes can meet the construction demand.
2- The boreholes must be secured with metal pipes and sealed both to increase the gas extraction efficiency and to avoid spontaneous combustion phenomena in the area of the boreholes.
Response: Thank you for the reviewer’s careful observation and very kind suggestion. We have carefully considered your comments and previous engineering practicals, and have changed the PVC material to metal materials in the revised manuscript.
Reviewer 2 Report
Based on the actual mining situation of the Liyazhuang coal mine, the detailed layout parameters of high-level boreholes were put forward by numerical simulation and field test. These have been applied successfully in engineering practice. The article's reasonable structure, purpose, and conclusions are clear. Some problems should be considered.
1. The pressure relief gas migration pathways in the vertical direction shown in section 3.2 (Figure 4) is 17 m, which is inconsistent with the description in this paper.
2. You must pay attention to your grammar and formatting problems, such as line 113 reference [17], please correct them in the paper.
3. Please check the presentation of the first sentence in 2.2 (2).
4. In the summary of 2.2, you need to show the high-level boreholes layout requirements that are mentioned above。
5. Figure 7 shows that the gas volume fraction at the upper corner decreases obviously under different extraction schemes. The fact extraction scheme compared with the non-extraction scheme on the corner gas volume fraction decreased significantly, please correct your statement.
There is no obvious problem in the overall structure of the article, but some details need to be modified. The revised paper can be accepted.
Author Response
- The pressure relief gas migration pathways in the vertical direction shown in section 3.2 (Figure 4) is 17 m, which is inconsistent with the description in this paper.
Response: Thank you for the reviewer’s comment. We have modified the expression in the text to match the content in figure 4 in the revised manuscript.
- You must pay attention to your grammar and formatting problems, such as line 113 reference [17], please correct them in the paper.
Response: Thank you for the reviewer’s careful observation and kind reminder. We feel very sorry for such typographical errors and have carefully checked the full text during the revision process in the revised manuscript.
- Please check the presentation of the first sentence in 2.2 (2).
Response: Thank you for the reviewer’s careful observation and very kind reminder. Based on the reviewer’s comment, we corrected the relevant sentences in the revised manuscript.
- In the summary of 2.2, you need to show the high-level boreholes layout requirements that are mentioned above。
Response: Thank you for the reviewer’s professional comment. We have written all the relevant content involved in the summary in the revised manuscript.
- Figure 7 shows that the gas volume fraction at the upper corner decreases obviously under different extraction schemes. The fact extraction scheme compared with the non-extraction scheme on the corner gas volume fraction decreased significantly, please correct your statement.
Response: Thank you for the reviewer’s careful observation and kind reminder. We have modified the representation according to Figure 7 in the revised manuscript.
Reviewer 3 Report
This article is very interesting. The gas disaster is a serious problem in the field of coal mining. Based on numerical simulation and field engineering practice to solve the problem is a reasonable and feasible idea. The problem and solving methods are clear and rigorous. I hope that the article can continue to improve.
1. The high-level boreholes layout position is in the lateral high drainage roadway, you need to pay attention to the lateral high drainage roadway layout position that can meet the needs of two working face services.
2. I have noticed some grammar and expression problems in the article, for example, in the first sentence in 2.2 (2).
3. Considering the influence of mining disturbance, whether the sealing length and material meet the effective extraction during the mining of the working face.
Author Response
1. The high-level boreholes layout position is in the lateral high drainage roadway, you need to pay attention to the lateral high drainage roadway layout position that can meet the needs of two working face services.
Response: Thank you for the reviewer’s very professional and leading-edge comment. We have done previous studies on high drainage roadway layout parameters, in order to achieve more effective and accurate extraction, we will continue to study the critical layout parameters of the high-level boreholes.
2. I have noticed some grammar and expression problems in the article, for example, in the first sentence in 2.2 (2).
Response: Thank you for the reviewer’s careful observation. Based on the reviewer’s comment, we corrected the relevant sentences in the revised manuscript and check the text again.
3. Considering the influence of mining disturbance, whether the sealing length and material meet the effective extraction during the mining of the working face.
Response: Thank you for the reviewer’s professional comment and kind reminding. By monitoring the effect of borehole extraction throughout the entire extraction stage and analyzing the extraction data, the length and material of the hole can meet the requirements.
Reviewer 4 Report
My estimation for this paper is Minor Revision.

Author Response
- There are some grammar, spelling, and sentence structure errors. The authors should have this paper reviewed by a native English speaker or a professional editorial organization.
Response: Thank you for reviewer’s careful observation. Based on the reviewer’s comment, we corrected the relevant sentences in the revised manuscript and check the text again.
- The whole article is not the final version. There are a lot red modifying annotations in the manuscript leading it hard to read. It is best to present the final modified version.
Response: Thank you for reviewer’s kind reminding. We have submitted the final version.
- The lines and words of Figures 2-4 are not clear enough, which is suggested to be redrawn.
Response: Thank you for reviewer’s careful observation. We have redrawn the correlation diagram in the manuscript.